# Characterizing Data Point Vulnerability via Average-Case Robustness

**Tessa Han**[*1]                **Suraj Srinivas**[*1]                **Himabindu Lakkaraju**[1]

[1]Harvard University

## Abstract

Studying the robustness of machine learning models is important to ensure consistent model behaviour across real-world settings. To this end, adversarial robustness is a standard framework, which views robustness of predictions through a binary lens: either a worst-case adversarial misclassification exists in the local region around an input, or it does not. However, this binary perspective does not account for the degrees of vulnerability, as data points with a larger number of misclassified examples in their neighborhoods are more vulnerable. In this work, we consider a complementary framework for robustness, called average-case robustness, which measures the fraction of points in a local region that provides consistent predictions. However, computing this quantity is hard, as standard Monte Carlo approaches are inefficient especially for high-dimensional inputs. In this work, we propose the first analytical estimators for average-case robustness for multi-class classifiers. We show empirically that our estimators are accurate and efficient for standard deep learning models and demonstrate their usefulness for identifying vulnerable data points, as well as quantifying robustness bias of models. Overall, our tools provide a complementary view to robustness, improving our ability to characterize model behaviour.

## 1   INTRODUCTION

A desirable attribute of machine learning models is robustness to perturbations of input data. A popular notion of robustness is adversarial robustness, the ability of a model to maintain its prediction when presented with adversar-

*Equal contribution

ial perturbations, i.e., perturbations designed to cause the model to change its prediction. Although adversarial robustness identifies whether a misclassified example exists in a local region around an input, it fails to capture the degree of vulnerability of that example, indicated by the difficulty in finding an adversary. For example, if the model geometry is such that 99% of the local region around an example (say, point $A$) contains correctly classified examples, this makes it harder to find an adversarial example as compared to the case where only 1% of the local region (say, for point $B$) contains correctly classified examples, where even random perturbations may be misclassified. However, from the adversarial robustness perspective, the prediction at a point is declared either robust or not, and thus both points $A$ and $B$ are considered equally non-robust (see Figure 1 for an illustrative example). The ease of obtaining a misclassification, or *data point vulnerability*, is captured by another kind of robustness: *average-case robustness*, i.e., the fraction of points in a local region around an input for which the model provides consistent predictions. [*] If this fraction is less than one, then an adversarial perturbation exists. The smaller this fraction, the easier it is to find a misclassified example. While adversarial robustness is motivated by model security, average-case robustness is better suited for model and dataset understanding, and debugging.

Standard approaches to computing average-case robustness involve Monte-Carlo sampling, which is computationally inefficient especially for high-dimensional data. For example, Cohen et al. [2019] use $n = 100,000$ Monte Carlo samples per data point to compute this quantity. In this paper, we propose to compute average-case robustness via

[*]In addition to the size of the misclassified region, another factor that affects the ease of finding misclassified examples is the specific optimization method used. In this study, we aim to study model robustness in a manner agnostic to the specific optimization used, and thus, we only focus on the size of the misclassified region. We believe this study can form the basis for future studies looking into the properties (e.g., "ease of identifying misclassified examples") of specific optimization methods.

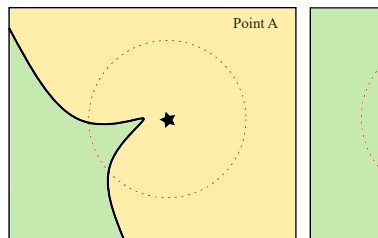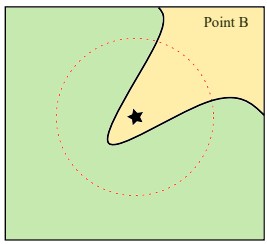

Figure 1: Consider a binary classifier (green vs. yellow) and points $A$ (left) and $B$ (right), both correctly classified to the yellow class. The dotted red circles represent $\epsilon$-balls around the data points. Although adversarial robustness rightly considers the model non-robust at both points (due to the existence of adversarial examples within the $\epsilon$-ball), it fails to discern that point $B$ has a larger fraction of misclassified points in its neighborhood, making it more vulnerable than point $A$, an aspect exactly captured by average-case robustness.

analytical estimators, reducing the computational burden, while simultaneously providing insight into model decision boundaries. Our estimators are exact for linear models and well-approximated for non-linear models, especially those having a small local curvature [Moosavi-Dezfooli et al., 2019, Srinivas et al., 2022]. Overall, our work makes the following contributions:

1. We derive novel analytical estimators to efficiently compute the average-case robustness of multi-class classifiers. We also provide estimation error bounds for these estimators that characterizes approximation errors for non-linear models.

2. We empirically validate our analytical estimators on standard deep learning models and datasets, demonstrating that these estimators accurately and efficiently estimate average-case robustness.

3. We demonstrate the usefulness of our estimators in two case studies: identifying vulnerable samples in a dataset and measuring class-level robustness bias [Nanda et al., 2021], where we find that standard models exhibit significant robustness bias among classes.

To our knowledge, this work is the first to investigate analytical estimation of average-case robustness for the multi-class setting. In addition, the efficiency of these estimators makes the computation of average-case robustness practical, especially for large deep neural networks.

## 2 RELATED WORK

**Adversarial robustness.** Prior works have proposed methods to generate adversarial attacks [Carlini and Wagner, 2017, Goodfellow et al., 2015, Moosavi-Dezfooli et al.,

2016], which find adversarial perturbations in a local region around a point. In contrast, this work investigates average-case robustness, which calculates the probability that a model's prediction remains consistent in a local region around a point. Prior works have also proposed methods to certify model robustness [Cohen et al., 2019, Carlini et al., 2022], guaranteeing the lack of adversarial examples for a given $\epsilon$-ball under certain settings. Specifically, Cohen et al. [2019] propose randomized smoothing, which involves computing class-wise average-case robustness, which is taken as the output probabilities of the randomized smoothed model. However, they estimate these probabilities via Monte Carlo sampling with $n = 100,000$ samples, which is computationally expensive. Viewing average-case robustness from the lens of randomized smoothing, our estimators can also be seen as providing an analytical estimate of randomized smoothed models. However, in this work, we focus on their applications for model understanding and debugging as opposed to improving robustness.

**Probabilistic robustness.** Prior works have explored notions of probabilistic and average-case robustness. For instance, Fazlyab et al. [2019], Kumar et al. [2020], Mangal et al. [2019] focus on certifying robustness of real-valued outputs to input perturbations. In contrast, this work focuses on only those output changes that cause misclassification. Like our work, Franceschi et al. [2018] also considers misclassifications. However, Franceschi et al. [2018] aims to find the smallest neighborhood with no adversarial example, while we compute the probability of misclassification in a given neighborhood. Robey et al. [2022], Rice et al. [2021] also aim to compute average-case robustness. However, they do so by computing the average loss over the neighborhood, while we use the misclassification rate. Closest to our work is the work by Weng et al. [2019] which aims to certify the binary misclassification rate (with respect to a specific class to misclassify to) using lower and upper linear bounds. In contrast, our work estimates the multi-class misclassification rate, as opposed to bounding the quantity in a binary setting. A crucial contribution of our work is its applicability to multi-class classification and the emphasis on estimating, rather than bounding, robustness.

**Robustness to distributional shifts.** Prior works have explored the performance of models under various distributions shifts [Taori et al., 2020, Ovadia et al., 2019]. From the perspective of distribution shift, average-case robustness can be seen as a measure of model performance under Gaussian noise, a type of natural distribution shift. In addition, in contrast to works in distributional robustness which seek to build models that are robust to distributions shifts [Thulasidasan et al., 2021, Moayeri et al., 2022], this work focuses on measuring the vulnerability of existing models to Gaussian distribution shifts.

# 3 AVERAGE-CASE ROBUSTNESS ESTIMATION

In this section, we first describe the mathematical problem of average-case robustness estimation. Then, we present the naïve estimator based on Monte Carlo sampling and derive more efficient analytical estimators.

## 3.1 NOTATION AND PRELIMINARIES

Assume that we have a neural network $f : \mathbb{R}^d \rightarrow \mathbb{R}^C$ with $C$ output classes and that the classifier predicts class $t \in [1,...,C]$ for a given input $\mathbf{x} \in \mathbb{R}^d$, i.e., $t = \arg\max_{i=1}^{C} f_i(\mathbf{x})$, where $f_i$ denotes the logits for the $i^{th}$ class. Given this classifier, the average-case robustness estimation problem is to compute the probability of consistent classification (to class $t$) under noise perturbation of the input.

**Definition 1.** *We define the **average-case robustness** of a classifier $f$ to noise $\mathcal{R}$ at a point $\mathbf{x}$ as*

$$p^{\text{robust}}(\mathbf{x}, t) = P_{\epsilon \sim R}\left[\arg\max_i f_i(x + \epsilon) = t\right]$$

The more robust the model is in the local neighborhood around $\mathbf{x}$, the larger the average-case robustness measure $p^{\text{robust}}(\mathbf{x}, t)$. In this paper, given that robustness is always measured with respect to the predicted class $t$ at $\mathbf{x}$, we henceforth suppress the dependence on $t$ in the notation. We also explicitly show the dependence of $p^{\text{robust}}$ on the noise scale $\sigma$ by denoting it as $p_\sigma^{\text{robust}}$.

In this work, we shall consider $\mathcal{R}$ as an isotropic Normal distribution, i.e., $\mathcal{R} = \mathcal{N}(0, \sigma^2)$. However, as we shall discuss in the next section, it is possible to accommodate both non-isotropic and non-Gaussian distributions in our method. Note that for high-dimensional data ($d \rightarrow \infty$), the isotropic Gaussian distribution converges to the uniform distribution on the surface of the sphere with radius $r = \sigma\sqrt{d}$ [†] due to the concentration of measure phenomenon [Vershynin, 2018].

Observe that when the domain of the input noise is restricted to an $\ell_p$ ball, $p_\sigma^{\text{robust}}$ generalizes the corresponding $\ell_p$ adversarial robustness. In other words, adversarial robustness is concerned with the quantity $\mathbf{1}(p_\sigma^{\text{robust}} < 1)$, i.e., the indicator function that average-case robustness is less than one (which indicates the presence of an adversarial perturbation), while this work focuses on computing the quantity $p_\sigma^{\text{robust}}$ itself. In the rest of this section, we derive estimators for $p_\sigma^{\text{robust}}$.

**The Monte-Carlo estimator.** A naïve estimator of average-case robustness is the Monte-Carlo estimator $p_\sigma^{\text{mc}}$. It computes the robustness of a classifier $f$ at input $\mathbf{x}$ by

---

[†]Alternately, if $\epsilon \sim \mathcal{N}(0, \sigma^2/d)$, then $r = \sigma$

generating $M$ noisy samples of $\mathbf{x}$ and then calculating the fraction of these noisy samples that are classified to the same class as $\mathbf{x}$. In other words,

$$
\begin{aligned}
p_\sigma^{\text{robust}}(\mathbf{x}) &= P_{\epsilon \sim \mathcal{N}(0,\sigma^2)}\left[\arg\max_i f_i(\mathbf{x}+\epsilon) = t\right] \\
&= \mathbb{E}_{\epsilon \sim \mathcal{N}(0,\sigma^2)}\left[\mathbf{1}_{\arg\max_i f_i(\mathbf{x}+\epsilon)=t}\right] \\
&\approx \frac{1}{M}\sum_{j=1}^{M}\left[\mathbf{1}_{\arg\max_i f_i(\mathbf{x}+\epsilon_j)=t}\right] = p_\sigma^{\text{mc}}(\mathbf{x})
\end{aligned}
$$

$p_\sigma^{\text{mc}}$ replaces the expectation with the sample average of the $M$ noisy samples of $\mathbf{x}$ and has been used in prior work [Nanda et al., 2021]. Technically, the error for the Monte-Carlo estimator is independent of dimensionality and is given by $\mathcal{O}(1/\sqrt{M})$ [Vershynin, 2018]. However, in practice, for neural networks, $p_\sigma^{\text{mc}}$ requires a large number of random samples to converge to the underlying expectation. For example, for MNIST and CIFAR10 CNNs, it takes around $M = 10,000$ samples per point for $p_\sigma^{\text{mc}}$ to converge, which is computationally expensive, and further, provides little information regarding the decision boundaries of the underlying model. Thus, we set out to address this problem by developing more efficient and informative analytical estimators of average-case robustness.

## 3.2 ROBUSTNESS ESTIMATION VIA LINEARIZATION

Before deriving analytical robustness estimators for non-linear models, we first consider the simpler problem of deriving this quantity for linear models. This is challenging, especially for multi-class classifiers. For example, given a linear model for a three-class classification problem with weights $w_1, w_2, w_3$ and biases $b_1, b_2, b_3$, such that $y = \arg\max_i\{w_i^\top \mathbf{x} + b_i \mid i \in [1,2,3]\}$, the decision boundary function between classes $i$ and $j$ is given by $y_{ij} = (w_i - w_j)^\top \mathbf{x} + (b_i - b_j)$. If the predicted label at $\mathbf{x}$ is $y = 1$, the relevant decision boundary functions are $y_{12}, y_{13}$ which characterize the decision boundaries of misclassifications from class 1 to classes 2, 3 respectively. To compute the total probability of misclassification, we must compute the probability of decision boundaries $y_{12}, y_{13}$ being crossed separately. Crucially, it is important not to "double count" the probability of both $y_{12}$ and $y_{13}$ being simultaneously crossed. Computing the probability of falling into this problematic region is non-trivial, as it depends on the relative orientations of $y_{12}$ and $y_{13}$. If they are orthogonal, then this problem is avoided, as the probability of crossing $y_{12}$ and $y_{13}$ are independent random variables. However, this is not true in general for non-orthogonal decision boundaries. Further, this "double counting" problem increases in complexity with an increasing number of classes, stemming from a corresponding increase in the number of such pairwise decision

boundaries. Lemma 1 provides an elegant solution to this combinatorial problem via the multivariate Gaussian CDF.

**Notation**: For clarity, we represent tensors by collapsing along the "class" dimension, i.e., $a_i \big|_{i=1}^C := (a_1, a_2, ...a_i, ...a_c)$, where for an order-$t$ tensor $a_i$, the expansion $a_i \big|_{i=1}^C$ is an order-$(t+1)$ tensor.

**Lemma 1.** *The local robustness of a multi-class linear model $f(\mathbf{x}) = \mathbf{w}^\top \mathbf{x} + b$ (with $\mathbf{w} \in \mathbb{R}^{d \times C}$ and $b \in \mathbb{R}^C$) at point $\mathbf{x}$ with respect to a target class $t$ is given by the following. Define weights $\mathbf{u}_i = \mathbf{w}_t - \mathbf{w}_i \in \mathbb{R}^d, \forall i \neq t$, where $\mathbf{w}_t, \mathbf{w}_i$ are rows of $\mathbf{w}$ and biases $c_i = \mathbf{u}_i^\top \mathbf{x} + (b_t - b_i) \in \mathbb{R}$. Then,*

$$p_\sigma^{\text{robust}}(\mathbf{x}) = \Phi_{\mathbf{U}\mathbf{U}^\top} \left( \frac{c_i}{\sigma \|\mathbf{u}_i\|_2} \Big|_{\substack{i=1 \\ i \neq t}}^C \right)$$

$$\text{where } \mathbf{U} = \frac{\mathbf{u}_i}{\|\mathbf{u}_i\|_2} \Big|_{\substack{i=1 \\ i \neq t}}^C \in \mathbb{R}^{(C-1) \times d}$$

*and $\Phi_{\mathbf{U}\mathbf{U}^\top}$ is the $(C-1)$-dimensional Normal CDF with zero mean and covariance $\mathbf{U}\mathbf{U}^\top$.*

*Proof Idea.* The proof involves constructing decision boundary functions $g_i(\mathbf{x}) = f_t(\mathbf{x}) - f_i(\mathbf{x})$ and computing the probability $p_\sigma^{\text{robust}}(\mathbf{x}) = P_\epsilon(\bigcup_{\substack{i=1 \\ i \neq t}}^C g_i(\mathbf{x} + \epsilon) > 0)$. For Gaussian $\epsilon$, we observe that $\frac{\mathbf{u}}{\sigma \|\mathbf{u}\|_2}^\top \epsilon \sim \mathcal{N}(0,1)$ is also a Gaussian, which applied vectorially results in our usage of $\Phi$. As convention, we represent $\mathbf{U}$ in a normalized form to ensure that its rows are unit norm. $\square$

The proof is in Appendix A.1. Thus, the multivariate Gaussian CDF provides an elegant solution to the previously mentioned "double counting" problem. Here, the matrix $\mathbf{U}$ exactly captures the linear decision boundaries, and the covariance matrix $\mathbf{U}\mathbf{U}^\top$ encodes the alignment between pairs of decision boundaries of different classes.

**Remark.** For the binary classification case, we get $\mathbf{U}\mathbf{U}^\top = 1$ (a scalar), and $p_\sigma^{\text{robust}}(\mathbf{x}) = \phi(\frac{c}{\sigma \|\mathbf{u}\|_2})$, where $\phi$ is the CDF of the scalar standard normal, which was previously also shown by Weng et al. [2019], Pawelczyk et al. [2023]. Hence Lemma 1 is a multi-class generalization of these works.

If the decision boundary vectors $\mathbf{u}_i$ are all orthogonal to each other, then the covariance matrix $\mathbf{U}\mathbf{U}^\top$ is the identity matrix. For diagonal covariance matrices, the multivariate Normal CDF (*mvn-cdf*) can be written as the product of univariate Normal CDFs, which is easy to compute. However, in practice, we find that the covariance matrix is strongly non-diagonal, indicating that the decision boundaries are not orthogonal to each other. This non-diagonal nature of covariance matrices in practice leads to the resulting *mvn-cdf* not having a closed form solution, and thus needing to be approximated via sampling [Botev, 2017, Sci]. However, this

sampling is performed in the $(C-1)$-dimensional space as opposed to the $d$-dimensional space that $p_\sigma^{\text{mc}}$ samples from. In practice, for classification problems, we often have $C << d$, making sampling in $(C-1)$-dimensions more efficient. We would like to stress here that the expression in Lemma 1 represents the simplest expression to compute the average-case robustness: the usage of the multi-variate Gaussian CDF cannot be avoided due to the computational nature of this problem. We now discuss the applicability of Lemma 1 to non-Gaussian noise.

**Lemma 2.** *(**Application to non-Gaussian noise**) For high-dimensional data ($d \to \infty$), Lemma 1 generalizes to any coordinate-wise independent noise distribution that satisfies Lyapunov's condition.*

*Proof Idea.* Applying Lyupanov's central limit theorem [Patrick, 1995], given $\epsilon \sim \mathcal{R}$ is sampled from some distribution $\mathcal{R}$, we have $\frac{\mathbf{u}}{\sigma \|\mathbf{u}\|_2}^\top \epsilon = \sum_{j=1}^d \frac{\mathbf{u}_j}{\sigma \|\mathbf{u}\|_2} \epsilon_j \xrightarrow{d} \mathcal{N}(0,1)$, which holds as long as the sequence $\{\frac{\mathbf{u}_j}{\|\mathbf{u}\|_2} \epsilon_j\}$ are independent random variables and satisfy the Lyapunov condition, which encodes the fact that higher-order moments of such distributions progressively shrink. $\square$

Thus, as long as the input noise distribution is "well-behaved", the central limit theorem ensures that the distribution of high-dimensional dot products is Gaussian, thus motivating our use of the *mvn-cdf* more generally beyond Gaussian input perturbations. We note that it is also possible to easily generalize Lemma 1 to **non-isotropic** Gaussian perturbations with a covariance matrix $\mathcal{C}$, which only changes the form of the covariance matrix of the *mvn-cdf* from $\mathbf{U}\mathbf{U}^\top \to \mathbf{U}\mathcal{C}\mathbf{U}^\top$, which we elaborate in Appendix A.1. In the rest of this paper, we focus on the isotropic case.

### 3.2.1 Estimator 1: The Taylor Estimator

Using the estimator derived for multi-class linear models in Lemma 1, we now derive the Taylor estimator, a local robustness estimator for non-linear models.

**Definition 2.** *The **Taylor estimator** for the local robustness of a classifier $f$ at point $\mathbf{x}$ with respect to target class $t$ is given by linearizing $f$ around $\mathbf{x}$ using a first-order Taylor expansion, with decision boundaries $g_i(\mathbf{x}) = f_t(\mathbf{x}) - f_i(\mathbf{x})$, $\forall i \neq t$, leading to*

$$p_\sigma^{\text{taylor}}(\mathbf{x}) = \Phi_{\mathbf{U}\mathbf{U}^\top} \left( \frac{g_i(\mathbf{x})}{\sigma \|\nabla_\mathbf{x} g_i(\mathbf{x})\|_2} \Big|_{\substack{i=1 \\ i \neq t}}^C \right)$$

*with $\mathbf{U}$ and $\Phi$ defined as in the linear case.*

The proof is in Appendix A.1. It involves locally linearizing non-linear decision boundary functions $g_i(\mathbf{x})$ using a Taylor series expansion. We expect this estimator to have a small

error when the underlying model is well-approximated by a locally linear function in the local neighborhood. We formalize this intuition by computing the estimation error for a quadratic classifier.

**Proposition 1.** *The **estimation error** of the Taylor estimator for a classifier with a quadratic decision boundary $g_i(\mathbf{x}) = \mathbf{x}^\top A_i \mathbf{x} + \mathbf{u}_i^\top \mathbf{x} + c_i$ and positive-semidefinite $A_i$ is upper bounded by*

$$|p_\sigma^{\text{robust}}(\mathbf{x}) - p_\sigma^{\text{taylor}}(\mathbf{x})| \leq k\sigma^{C-1} \prod_{\substack{i=1 \\ i \neq t}}^{C} \frac{\lambda_{\max}^{A_i}}{\|\mathbf{u}_i\|_2}$$

*for noise $\epsilon \sim \mathcal{N}(0, \sigma^2/d)$, in the limit of $d \to \infty$. Here, $\lambda_{\max}^{A_i}$ is the max eigenvalue of $A_i$, and $k$ is a small problem dependent constant.*

The proof is in Appendix A.1. This statement formalizes two key intuitions with regards to the Taylor estimator: (1) the estimation error depends on the size of the local neighborhood $\sigma$ (the smaller the local neighborhood, the more locally linear the model, and the smaller the estimator error), and (2) the estimation error depends on the extent of non-linearity of the underlying function, which is given by the ratio of the max eigenvalue of $A$ to the Frobenius norm of the linear term. This measure of non-linearity of a function, called normalized curvature, has also been independently proposed by previous work [Srinivas et al., 2022]. Notably, if the max eigenvalue is zero, then the function $g_i(\mathbf{x})$ is exactly linear, and the estimation error is zero, reverting back to the linear case in Lemma 1.

### 3.2.2 Estimator 2: The MMSE Estimator

While the Taylor estimator is more efficient than the Monte Carlo estimator, it has a drawback: its linearization is only faithful at perturbations close to the data point and not necessarily for larger perturbations. To mitigate this issue, we use a form of linearization that is faithful over larger noise perturbations. Linearization has been studied in feature attribution research, which concerns itself with approximating non-linear models with linear ones to produce model explanations [Han et al., 2022]. In particular, the SmoothGrad [Smilkov et al., 2017] technique has been described as the MMSE (minimum mean-squared error) optimal linearization of the model [Han et al., 2022, Agarwal et al., 2021] in a Gaussian neighborhood around the data point. Using a similar idea, we propose the MMSE estimator $p_\sigma^{\text{mmse}}$ as follows.

**Definition 3.** *The **MMSE estimator** for the local robustness of a classifier $f$ at point $\mathbf{x}$ with respect to target class $t$ is given by an MMSE linearization $f$ around $\mathbf{x}$, for decision*

boundaries $g_i(\mathbf{x}) = f_t(\mathbf{x}) - f_i(\mathbf{x})$, $\forall i \neq t$, leading to

$$p_\sigma^{\text{mmse}}(\mathbf{x}) = \Phi_{\mathbf{UU}^\top} \left( \frac{\tilde{g}_i(\mathbf{x})}{\sigma\|\nabla_\mathbf{x} \tilde{g}_i(\mathbf{x})\|_2} \Big|_{\substack{i=1 \\ i \neq t}}^{C} \right)$$

where $\tilde{g}_i(\mathbf{x}) = \frac{1}{N} \sum_{j=1}^{N} g_i(\mathbf{x} + \epsilon)$, $\epsilon \sim \mathcal{N}(0, \sigma^2)$

*with $\mathbf{U}$ and $\Phi$ defined as in the linear case, and $N$ is the number of perturbations.*

The proof is in Appendix A.1. It involves creating a randomized smooth model [Cohen et al., 2019] from the base model and computing the decision boundaries of this smooth model. Note that this estimator also involves drawing noise samples like the Monte Carlo estimator. However, unlike the Monte Carlo estimator, we find that the MMSE estimator converges fast (around $N = 5$), leading to an empirical advantage. We now compute the estimation error of the MMSE estimator.

**Proposition 2.** *The **estimation error** of the MMSE estimator for a classifier with a quadratic decision boundary $g_i(\mathbf{x}) = \mathbf{x}^\top A_i \mathbf{x} + \mathbf{u}_i^\top \mathbf{x} + c_i$ and positive-semidefinite $A_i$ is upper bounded by*

$$|p_\sigma^{\text{robust}}(\mathbf{x}) - p_\sigma^{\text{mmse}}(\mathbf{x})| \leq k\sigma^{C-1} \prod_{\substack{i=1 \\ i \neq t}}^{C} \frac{\lambda_{\max}^{A_i} - \lambda_{mean}^{A_i}}{\|\mathbf{u}_i\|_2}$$

*for noise $\epsilon \sim \mathcal{N}(0, \sigma^2/d)$, in the limit of $d \to \infty$ and $N \to \infty$. Here, $\lambda_{\max}^{A_i}, \lambda_{mean}^{A_i}$ are the maximum and mean eigenvalue of $A_i$ respectively, and $k$ is a small problem dependent constant.*

The proof is in Appendix A.1. The result above highlights two aspects of the MMSE estimator: (1) it incurs a smaller estimation error than the Taylor estimator, and (2) even in the limit of large number of samples $N \to \infty$, the error of the MMSE estimator is non-zero, except when $\lambda_{\text{mean}}^{A_i} = \lambda_{\max}^{A_i}$. For PSD matrices, this becomes zero when $A_i$ is a multiple of the identity matrix [‡], reverting back to the linear case in Lemma 1.

### 3.2.3 (Optionally) Approximating *mvn-cdf*: Connecting Robustness Estimation with Softmax

**Approximation with Multivariate Sigmoid.** One drawback of the Taylor and MMSE estimators is their use of the *mvn-cdf*, which does not have a closed form solution and can cause the estimators to be slow for settings with a large number of classes $C$. In addition, the *mvn-cdf* makes

---

[‡] When $d \to \infty$, $\epsilon^\top A \epsilon = \lambda\|\epsilon\|^2 = \lambda\sigma^2$ is a constant, and thus an isotropic quadratic function resembles a linear one in this neighborhood.

these estimators non-differentiable, which is inconvenient for applications which require differentiating $p_\sigma^{\text{robust}}$. To alleviate these issues, we approximate the *mvn-cdf* with an analytical closed-form expression. As CDFs are monotonically increasing functions, the approximation should also be monotonically increasing.

To this end, it has been previously shown that the *univariate* Normal CDF $\phi$ is well-approximated by the sigmoid function [Hendrycks and Gimpel, 2016]. It is also known that when $\mathbf{U}\mathbf{U}^\top = I$, *mvn-cdf* is given by $\Phi(\mathbf{x}) = \prod_i \phi(\mathbf{x}_i)$, i.e., it is given by the product of the univariate normal CDFs. Thus, we may choose to approximate $\Phi(\mathbf{x}) = \prod_i \text{sigmoid}(\mathbf{x})$. However, when the inputs are small, this can be simplified as follows:

$$
\begin{aligned}
\Phi_I(\mathbf{x}) &= \prod_i \phi(\mathbf{x}_i) \approx \prod_i \frac{1}{1 + \exp(-\mathbf{x}_i)} \\
&= \frac{1}{1 + \sum_i \exp(-\mathbf{x}_i) + \sum_{j,k} \exp(-\mathbf{x}_j - \mathbf{x}_k) + ...} \\
&\approx \frac{1}{1 + \sum_i \exp(-\mathbf{x}_i)} \quad (\text{for } \mathbf{x}_i \to \infty \ \forall i)
\end{aligned}
$$

We call the final expression the "multivariate sigmoid" (*mv-sigmoid*) which serves as our approximation of *mvn-cdf*, especially at the tails of the distribution. While we expect estimators using *mv-sigmoid* to approximate ones using *mvn-cdf* only when $\mathbf{U}\mathbf{U}^\top = \mathbf{I}$, we find experimentally that the approximation works well even for practical values of the covariance matrix $\mathbf{U}\mathbf{U}^\top$. Using this approximation to substitute *mv-sigmoid* for *mvn-cdf* in the $p_\sigma^{\text{taylor}}$ and $p_\sigma^{\text{mmse}}$ estimators yields the $p_\sigma^{\text{taylor\_mvs}}$ and $p_\sigma^{\text{mmse\_mvs}}$ estimators, respectively. We present further analysis on the multivariate sigmoid in Appendix A.4.

**Approximation with Softmax.** A common method to estimate the confidence of model predictions is to use the softmax function applied to the logits $f_i(\mathbf{x})$ of a model. We note that softmax is identical to *mv-sigmoid* when directly applied to the logits of neural networks:

$$
\begin{aligned}
\text{softmax}_t\left(f_i(\mathbf{x})\ \Big|_{i=1}^C\right) &= \frac{\exp(f_t(\mathbf{x}))}{\sum_{i=1}^C \exp(f_i(\mathbf{x}))} = \\
\frac{1}{1 + \sum_{\substack{i=1 \\ i \neq t}}^C \exp(f_i(\mathbf{x}) - f_t(\mathbf{x}))} &= \text{mv-sigmoid}\left(g_i(\mathbf{x})\ \Big|_{\substack{i=1 \\ i \neq t}}^C\right)
\end{aligned}
$$

Recall that $g_i(\mathbf{x}) = f_t(\mathbf{x}) - f_i(\mathbf{x})$ is the decision boundary function. Note that this equivalence only holds for the specific case of logits, and cannot be applied to approximate the Taylor estimator, for instance. Nonetheless, given this

similarity, it is reasonable to ask whether softmax applied to logits (henceforth $p_T^{\text{softmax}}$ for softmax with temperature $T$) itself can be a "good enough" estimator of $p_\sigma^{\text{robust}}$ in practice. In other words, does $p_T^{\text{softmax}}$ well-approximate $p_\sigma^{\text{robust}}$ in certain settings? In Appendix A.1, we provide a theoretical result for a restricted linear setting where softmax can indeed match the behavior of $p_\sigma^{\text{taylor\_mvs}}$, which happens precisely when $\mathbf{U}\mathbf{U}^\top = \mathbf{I}$ and all the class-wise gradients are equal. In the next section, we demonstrate empirically that the softmax estimator $p_T^{\text{softmax}}$ is a poor estimator of average-case robustness in practice.

## 4 EMPIRICAL EVALUATION

In this section, we first evaluate the estimation errors and computational efficiency of the analytical estimators, and then evaluate the impact of robustness training within models on these estimation errors. Then, we analyze the relationship between average-case robustness and softmax probability. Lastly, we demonstrate the usefulness of local robustness for model and dataset understanding with two case studies. Key results are discussed in this section and full results are in Appendix A.4.

**Datasets and models.** We evaluate the estimators on four datasets: MNIST [Deng, 2012], FashionMNIST [Xiao et al., 2017], CIFAR10 [Krizhevsky et al., 2009], and CIFAR100 [Krizhevsky et al., 2009]. For MNIST and FashionMNIST, we train linear models and CNNs. For CIFAR10 and CIFAR100, we train Transformer models. We also train ResNet18 models [He et al., 2016] using varying levels of gradient norm regularization [Srinivas and Fleuret, 2018, Srinivas et al., 2024] to obtain models with varying levels of robustness. For gradient norm regularization, the objective function is $\ell(f(x), y) + \lambda \|\nabla_x f(x)\|_2^2$, where $\lambda$ is the regularization constant. The larger $\lambda$ is, the more robust the model. Note that gradient norm regularization is equivalent to Gaussian data augmentation with an infinite number of augmented samples [Srinivas and Fleuret, 2018] and is different from adversarial training. Unless otherwise noted, the experiments below use each dataset's test set which consists of 10,000 points. Additional details about the datasets and models are described in Appendix A.2 and A.3.

### 4.1 EVALUATION OF THE ESTIMATION ERRORS OF ANALYTICAL ESTIMATORS

**The analytical estimators accurately compute local robustness.** To empirically evaluate the estimation error of our estimators, we calculate $p_\sigma^{\text{robust}}$ for each model using $p_\sigma^{\text{mc}}$, $p_\sigma^{\text{taylor}}$, $p_\sigma^{\text{mmse}}$, $p_\sigma^{\text{taylor\_mvs}}$, $p_\sigma^{\text{mmse\_mvs}}$, and $p_T^{\text{softmax}}$ for different $\sigma$ values. For $p_\sigma^{\text{mc}}$, $p_\sigma^{\text{mmse}}$, and $p_\sigma^{\text{mmse\_mvs}}$, we use a sample size at which these estimators have converged ($n = 10000, 500$, and $500$, respectively). (Convergence analyses are in Appendix A.4.) We take the Monte-

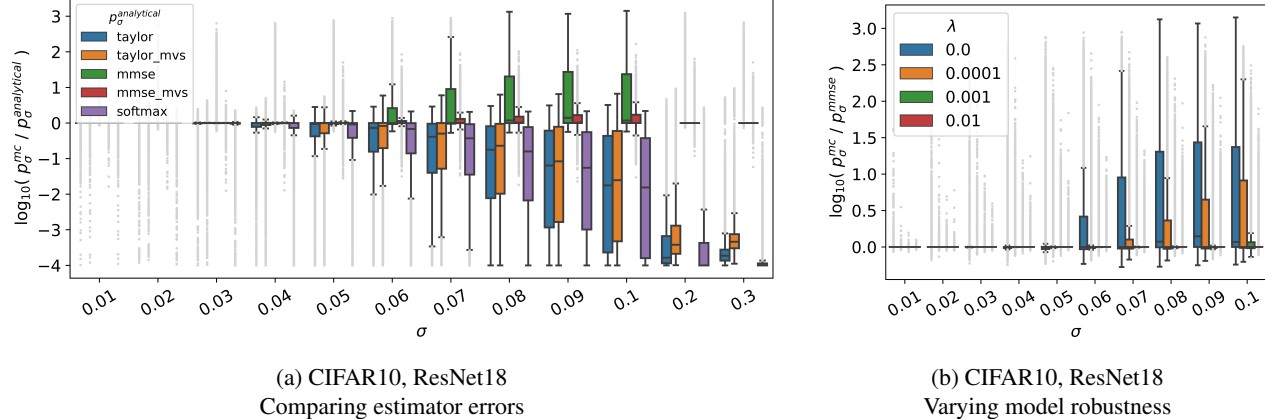

|  | (a) CIFAR10, ResNet18 | (b) CIFAR10, ResNet18 |
|---|---|---|
|  | Comparing estimator errors | Varying model robustness |

Figure 2: Empirical evaluation of analytical estimators. (a) The smaller the noise neighborhood $\sigma$, the more accurately the estimators compute $p_\sigma^{robust}$. $p_\sigma^{mmse}$ and $p_\sigma^{mmse\_mvs}$ are the best estimators of $p_\sigma^{robust}$, followed closely by $p_\sigma^{taylor\_mvs}$ and $p_\sigma^{taylor}$, trailed by $p_T^{softmax}$. (b) For more robust models, the estimators compute $p_\sigma^{robust}$ more accurately over a larger $\sigma$. Together, these results indicate that the analytical estimators accurately compute $p_\sigma^{robust}$.

Carlo estimator as the gold standard estimate of $p_\sigma^{robust}$), and compute the absolute and relative difference between $p_\sigma^{mc}$ and the other estimators to evaluate their estimation errors.

The performance of the estimators for the CIFAR10 ResNet18 model is shown in Figure 2a. The results indicate that $p_\sigma^{mmse\_mvs}$ and $p_\sigma^{mmse}$ are the best estimators of $p_\sigma^{robust}$, followed closely by $p_\sigma^{taylor\_mvs}$ and $p_\sigma^{taylor}$, trailed by $p_T^{softmax}$. This is consistent with the theory in Section 3, where the analytical estimation errors of $p_\sigma^{mmse}$ are lower than $p_\sigma^{taylor}$.

The results also confirm that the smaller the noise neighborhood $\sigma$, the more accurately the estimators compute $p_\sigma^{robust}$. For the MMSE and Taylor estimators, this is because their linear approximation of the model around the input is more faithful for smaller $\sigma$. As expected, when the model is linear, $p_\sigma^{taylor}$ and $p_\sigma^{mmse}$ accurately compute $p_\sigma^{robust}$ for all $\sigma$'s (Appendix A.4). For the softmax estimator, $p_T^{softmax}$ values are constant over $\sigma$ and this particular model has high $p_T^{softmax}$ values for most points. Thus, for small $\sigma$'s where $p_\sigma^{robust}$ is near one, $p_T^{softmax}$ happens to approximate $p_\sigma^{robust}$ for this model. Examples of images with varying levels of noise ($\sigma$) are in Appendix A.4.

**Impact of robust training on estimation errors.** The performance of $p_\sigma^{mmse}$ for CIFAR10 ResNet18 models of varying levels of robustness is shown in Figure 2b. The results indicate that the estimator is more accurate for more robust models (larger $\lambda$) over a larger $\sigma$. This is because robust training leads to models that are more locally linear [Moosavi-Dezfooli et al., 2019], making the estimator's linear approximation of the model around the input more accurate over a larger $\sigma$, making its $p_\sigma^{robust}$ values more accurate.

**Evaluating estimation error of mv-sigmoid.** To examine *mv-sigmoid*'s approximation of *mvn-cdf*, we compute both functions using the same inputs ($z = \frac{g_i(\mathbf{x})}{\sigma \|\nabla_\mathbf{x} g_i(\mathbf{x})\|_2}|_{\substack{i=1 \\ i \neq t}}^{C}$, as described in Proposition 2) for the CIFAR10 ResNet18 model for different $\sigma$. The plot of *mv-sigmoid(z)* against *mvn-cdf(z)* for $\sigma = 0.05$ is shown in Appendix A.4 (Figure 10). The results indicate that the two functions are strongly positively correlated with low approximation error, suggesting that *mv-sigmoid* approximates the *mvn-cdf* well in practice.

## 4.2 EVALUATION OF COMPUTATIONAL EFFICIENCY OF ANALYTICAL ESTIMATORS

**The analytical estimators are more efficient than the naïve estimator.** We examine the efficiency of the estimators by measuring their runtimes when calculating $p_{\sigma=0.1}^{robust}$ for the CIFAR10 ResNet18 model for 50 points. Runtimes are displayed in Table 1. They indicate that $p_\sigma^{taylor}$ and $p_\sigma^{mmse}$ perform 35x and 17x faster than $p_\sigma^{mc}$, respectively. Additional runtimes are in Appendix A.4.

We also examine the efficiency of the analytical estimators in terms of memory usage. The backward pass is observed to take about twice the amount of floating-point operations (FLOPs) as a forward pass [flo]. In addition, we performed an experiment and found that a forward and backward pass uses about twice the peak memory of a single forward pass. Thus, each iteration of $p_\sigma^{mmse}$ (which consists of a forward and backward pass) is roughly 3x the number of FLOPs and twice the peak memory of a single iteration of $p_\sigma^{mc}$ (which consists of one forward pass). However, $p_\sigma^{mmse}$ requires 5 iterations for convergence while $p_\sigma^{mc}$ requires about 10,000. Thus, overall, $p_\sigma^{mmse}$ is more memory-efficient than $p_\sigma^{mc}$.

| Estimator | Number of Samples ($n$) | CPU Runtime (h:m:s) | GPU Runtime (h:m:s) |
|---|---|---|---|
| $p_\sigma^{\text{mc}}$ | $n = 10{,}000$ | 1:41:11 | 0:19:56 |
| $p_\sigma^{\text{taylor}}$ | N/A | 0:00:08 | 0:00:02 |
| $p_\sigma^{\text{mmse}}$ | $n = 5$ | 0:00:41 | 0:00:06 |

Table 1: Runtimes of $p_\sigma^{\text{robust}}$ estimators. Each estimator computes $p_{\sigma=0.1}^{\text{robust}}$ for the CIFAR10 ResNet18 model for 50 data points. Estimators that use sampling use the minimum number of samples necessary for convergence. Runtimes are in the format of hour:minute:second. The GPU used was a Tesla V100. The analytical estimators ($p_\sigma^{\text{taylor}}$ and $p_\sigma^{\text{mmse}}$) are more efficient than the naïve estimator ($p_\sigma^{\text{mc}}$).

### 4.3 CASE STUDIES

**Identifying non-robust data points.** While robustness is typically viewed as the property of a model, the average-case robustness perspective compels us to view robustness as a joint property of both the model and the data point. In light of this, we can ask, given the same model, which samples are robustly and non-robustly classified? We evaluate whether $p_\sigma^{\text{robust}}$ can distinguish such images better than $p_T^{\text{softmax}}$. To this end, we train a simple CNN to distinguish between images with high and low $p_\sigma^{\text{mmse}}$ and the same CNN to also distinguish between images with high and low $p_T^{\text{softmax}}$ (additional setup details described in Appendix A.4). Then, we compare the performance of the two models. For CIFAR10, the test set accuracy for the $p_\sigma^{\text{mmse}}$ CNN is **92%** while that for the $p_T^{\text{softmax}}$ CNN is **58%**. These results indicate that $p_\sigma^{\text{robust}}$ better identifies images that are robust to and vulnerable to random noise than $p_T^{\text{softmax}}$.

We also present visualizations of images with the highest and lowest $p_\sigma^{\text{mmse}}$ in each class for each model. For comparison, we do the same with $p_T^{\text{softmax}}$. Example CIFAR10 images are shown in Figure 3. We observe that images with low $p_\sigma^{\text{robust}}$ tend to have neutral colors, with the object being a similar color as the background (making the prediction likely to change when the image is slightly perturbed), while images with high $p_\sigma^{\text{robust}}$ tend to be brightly-colored, with the object strongly contrasting with the background (making the prediction likely to stay constant when the image is slightly perturbed). Recall that points with small $p_\sigma^{\text{robust}}$ are close to the decision boundary, while those farther away have a high $p_\sigma^{\text{robust}}$. Thus, high $p_\sigma^{\text{robust}}$ points may be thought of as "canonical" examples of the underlying class, while low $p_\sigma^{\text{robust}}$ examples are analogous to "support vectors", that are critical to model learning. These results showcase the utility of average-case robustness for dataset exploration and analysis, particularly in identifying canonical and outlier examples.

**Detecting robustness bias among classes: Is the model differently robust for different classes?** We also demon-

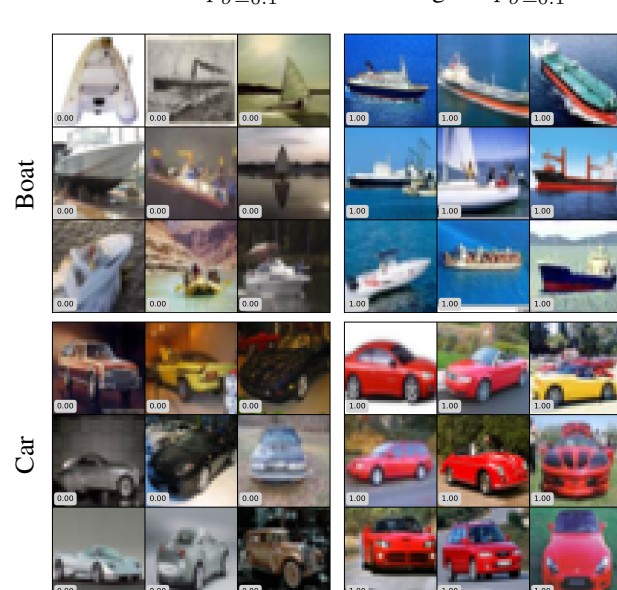

Figure 3: Example ranking of $p_\sigma^{\text{robust}}$ among CIFAR10 classes. Images with high $p_\sigma^{\text{robust}}$ are farther away from the decision boundary, and tend to be brighter and have stronger object-background contrast than those with low $p_\sigma^{\text{robust}}$, which are closer to the decision boundary, and thus easily misclassified.

strate that $p_\sigma^{\text{robust}}$ can detect bias in local robustness [Nanda et al., 2021] by examining its distribution for each class for each model and test set over different $\sigma$'s. Results for the CIFAR10 ResNet18 model are in plotted in Figure 4. The results show that different classes have significantly different $p_\sigma^{\text{robust}}$ distributions, i.e., the model is significantly more robust for some classes (e.g., frog) than for others (e.g., airplane). Similarly for the FMNIST CNN case in Figure 4, we find that the pullover class is much less robust than the sandal class. This observation indicates a disparity in outcomes for these different classes, and underscores the importance of evaluating per-class and per-datapoint robustness metrics before deploying models in the wild. The results also show that $p_\sigma^{\text{mc}}$ and $p_\sigma^{\text{mmse}}$ have very similar distributions, further indicating that the latter well-approximates the former. $p_\sigma^{\text{robust}}$ detects robustness bias across all other models and datasets too: MNIST CNN, and CIFAR100 ResNet18 (Appendix A.4). Thus, $p_\sigma^{\text{robust}}$ can be applied to detect robustness bias among classes, which is critical when models are deployed in high-stakes, real-world settings.

## 5 CONCLUSION

In this work, we find that adversarial robustness does not provide a comprehensive picture of model behavior, and to this end, we propose the usage of average-case robustness. While adversarial robustness is more suited for applica-

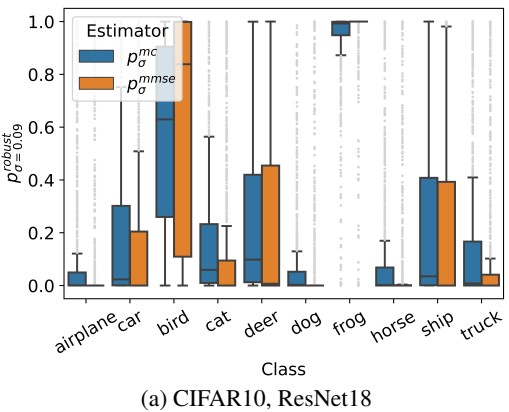

(a) CIFAR10, ResNet18

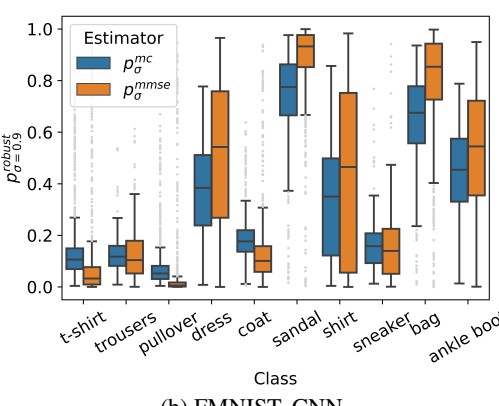

(b) FMNIST, CNN

Figure 4: Computing robustness bias among classes for the (a) ResNet18 CIFAR10 model, and (b) for the CNN FMNIST model. $p_\sigma^{\text{robust}}$ reveals that the model robustness varies significantly across classes, revealing a marked class-wise bias within standard models. The analytical estimator $p_\sigma^{\text{mmse}}$ accurately captures this model bias.

tions in model security, average-case robustness is suited for model understanding and debugging. To our knowledge, this work is the first to investigate analytical estimators for average-case robustness in a multi-class setting. The analytical aspect of these estimators helps understand average-case robustness via model decision boundaries, and also connect to ideas such as Randomized Smoothing (via the MMSE estimator) and softmax probabilities.

Future research directions include exploring additional applications of average-case robustness, such as training average-case robust models that minimize the probability of misclassification and debugging-oriented applications such as detecting model memorization, and dataset outliers.

**Code availability.** Code is available at `https://github.com/AI4LIFE-GROUP/average-case-robustness`.

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
