# OpenReview forum: "Characterizing Data Point Vulnerability as Average-Case Robustness"
_auai.org/UAI/2024/Conference — UAI 2024 poster_

### Official Review · Reviewer_PM6A · 2024-03-11

**Q2-1 Originality-Novelty:** 3
**Q2-2 Correctness-Technical Quality:** 3
**Q2-5 Clarity Of Writing:** 4

**Q1 Summary And Contributions:**

The paper proposes to investigate the adversarial robustness through the lens of average-case perturbations, instead of the worst-case perturbations. Then, the paper proposes two efficient Monte Carlo estimators. Theoretical bounds of estimation errors are derived for both estimators. The estimators are evaluated empirically on (Fashion)MNIST and CIFAR-10(0).

**Q2-3 Extent To Which Claims Are Supported By Evidence:**

2: Fair: the main claims are somewhat supported by evidence (but the experimental evaluation may be weak, or does not match entirely with the claims, important baselines may be missing, proofs contain important ideas but lack rigor, algorithmic details are only discussed superficially, references are imprecise, assumptions are not sufficiently motivated or explicated, etc.).

**Q2-4 Reproducibility:**

2: Fair: key resources (e.g. proofs, code, data) are unavailable but key details (e.g. proof sketches, experimental setup) are sufficiently well-described for an expert to confidently reproduce the main results.

**Q3 Main Strengths:**

- **Theoretical analysis**. The analytical derivations of the bounds of the estimation errors are strong. The paper built solid theoretical foundations for its technical contributions.
- **Impact**. The paper proposes an effective solution that may impact several problems (which should be better defined), like randomized smoothing.
- **Novelty**. The paper provides valuable non-trivial technical contributions (however, the evaluation of novelty of the current paper is hard, see comments below).
- **Writing**. The paper is clear and well-written.

**Q4 Main Weakness:**

- No mention of distributional shift (Novelty, related work and motivation)
  - Weak motivation
  - Related work
- Weak experimental design
  - Limited experimental design
  - Weak experimental design of the robust training experiment
  - Weak Case Studies

See details in Q5.

**Q5 Detailed Comments To The Authors:**

- **Distributional shift (Novelty, related work and motivation)**. The main issue of the paper is the lack of discussion of natural (as opposed to adversarial) distribution shift. This is surprising because the paper is dedicated to the robustness under Gaussian noise, which is the canonical distributional shift. The paper states that its study of the adversarial perturbations using the notion of average-case perturbation is novel. The issue is that this notion makes little sense from an adversarial perspective, since the adversarial nature implies worst-case distributional shift. Consider the average-case as part of adversarial machine learning is weak. For this reason, the motivation of the paper is weak. I am surprised that the related work and background have no mention of (natural) distributional shift. This weakness makes it difficult to position the novelty of the paper, since the most related work and adequate background is not addressed.
   - **Weak motivation**. As previously stated, motivating the robustness to Gaussian noise through the lens of adversarial ML is not convincing. The paper uses vague motivations, like model debugging and model understanding, without providing details and precise justification. The connection with random smoothing is much more promising, but is just briefly mentioned in related work and neither derived nor evaluated. The paper would greatly benefit from a stronger connection to random smoothing. For example, the paper could evaluate different random smoothing techniques with their strategy and evaluate the certified robustness.
   - **Related work**. As previously stated, the most related work is missing. The paper should discuss extensive existing work about robustness to Gaussian noise.
- **Weak experimental design**
  - **Limited experimental design**. It would be nice to run the experiments on larger dataset than CIFAR10(0). Since the technique can use available pretrained model, I believe that the computational cost of the experiments on ImageNet is feasible. Moreover, the paper would be strengthen by evaluating other architectures on CIFAR-10(0) than ResNet-18 only. The study is limited to small and outdated architectures, with a lack of diversity and contemporary architectures such as MLP or Swin Transformers. This limited selection hampers the generalization of the findings.
  - **Weak experimental design of the robust training experiment**. In Section 4.1 the paper claims to study the impact of robust training, but the paper actually studies the impact of the weight decay hyperparameter. The word "weight decay" is not mentioned, instead "robust training" and "gradient norm regularization" are used. This is confusing for three reasons: (i) weight decay regularization does not control the level of robustness, at least not directly (ii) several gradient norm regularizers exist, so the wording is ambiguous (iii) robust training can be mistaken for adversarial training. Since the weight decay does not control directly the robustness to Gaussian noise, it would be more useful to evaluate different training techniques designed to increase the robustness to distribution shift. At least, the wording needs to change. Ideally, the evaluation protocol would be improved to include evaluate training techniques designed to improve the robustness to distribution shift.
  - **Weak Case Studies**. The first case study is weak because unrelated to existing techniques. Identifying non-robust data points seems to be very connected to the notion of uncertainty estimation: differentiating certain and uncertain predictions. But the notion of uncertainty is not mentioned. The field of uncertainty is well studied, so it is unclear whether the paper provides a novel approach to this problem. It would be great to compare the approach with existing uncertainty estimation techniques. The experimental design of this experiment is also unclear and non-standard. Training a CNN to distinguish between high and low $p\_\\sigma^{mmse}$ seems inappropriate. A standard benchmark of uncertainty estimation would be more appropriate. Figure 3 and the associated comment seem to describe the notion of hard examples, without mentioning it.

### Minor points

- **Computational cost**. Section 3.2 states that the two proposed estimators converge in less iterations than its baseline. It would be nice to briefly discuss the computational cost of each iteration. For example, MMSE needs to compute input gradients, which doubles the computational cost and requires more memory. As shown in Table 1, the proposed estimators are still more efficient than the MC baseline, due to several order of magnitude less iterations needed. But it would be better to discuss cost per iteration when mentioning the number of iterations.
- **Presentation**: Figure 2 is hard to read and interpret. The estimation error for low sigma is impossible to see. It may be better to report relative deviations from $p\_\\sigma^{mc}$.
- **Notation**. The use of epsilon for the Gaussian random variable can be confusing. In the adversarial example literature, this notation is generally reserved for the size of the L-p ball.

### Questions for authors


- In Section 3.2.3 (bottom right of page 5), why would $x_i \\rightarrow\\infty \\forall i$?
- Will the source code of the experiments be publicly available?

**Q9 Complying With Reviewing Instructions:**

Yes

---

> ### Author Rebuttal · Authors · 2024-04-04
>
> Dear Reviewer PM6A,
>
> Thank you! We appreciate your review and are glad you liked the paper overall! We address your comments below.
>
> > Discuss natural distribution shift
>
> Thanks for this comment! Indeed, average-case robustness measures model performance under Gaussian noise, a type of natural distribution shift. In the paper, we discussed works considering robustness to such noise (Section 2, under “Probabilistic robustness”). While the field of “distributional robustness” (which considers building models robust to distribution shifts) is related, this work focuses on measuring the vulnerability of existing models to Gaussian distribution shifts. We will add this discussion of distributional robustness to the paper.
>
> > “vague motivations, like model debugging and model understanding”
>
> We concretize model debugging and understanding in our experiments (Section 4), where we demonstrate how average-case robustness helps with model understanding (e.g., identifying non-robust points; Fig. 3) and model debugging (e.g., detecting robustness bias; Fig. 4). We agree that model understanding and debugging do not constitute a single use-case but are rather umbrella terms for methods providing insight into model behavior, and in this case, model decision boundaries.
>
> > Add “stronger connection to randomized smoothing”
>
> Randomized smoothing certifies model robustness by first computing a quantity equivalent to p_robust using Monte-Carlo estimation. In contrast, our work focuses on computing p_robust more efficiently using analytical estimators. Consequently, in randomized smoothing, p_robust is part of the model design (it is the model’s prediction). In contrast, our work uses p_robust after a model is trained as a post hoc method to measure model robustness. However, given this connection, we think it can be an interesting direction to build empirically robust models using our analytical estimators instead of Monte Carlo estimators in randomized smoothing.
>
> > ImageNet experiments
>
> Thanks for the comment! One computational bottleneck for experiments is the baseline p_mc, which is intractable for large models and datasets. Another bottleneck is the number of classes (more so than the model and dataset size). For example, calculating p_robust analytically for ImageNet (1000 classes) requires computing a 999-dimensional MVN CDF, which is intractable. This is due to the inherent complexity of the underlying mathematical object (mvn-cdf), which does not have a closed-form solution. However, we hope to extend our work to large-scale models and datasets in future work.
>
> > “the paper actually studies the impact of the weight decay hyperparameter”
>
> We believe the reviewer may have misunderstood this aspect. We do not use weight decay. We use gradient norm regularization to train robust models. The objective function is $l(f(x), y) + \lambda || \nabla_x f(x) ||_2^2$ where l(f(x),y) is cross-entropy loss, $\lambda$ is the regularization constant, and $f(x)$ is the largest softmax probability for input $x$. Gradient/Jacobian-based regularization helps improve model robustness, as shown in prior works (E.g.: Hoffman et al, “Robust Learning with Jacobian Regularization”, arxiv 2019).
>
> > Connection to uncertainty estimation
>
> Good point! While it intuitively seems that p_robust is connected to uncertainty estimation, in practice we did not find a clear connection to typical uncertainty estimation metrics like calibration (i.e., likelihood of misclassification). However, we did find p_robust to accurately capture the likelihood of misclassification **upon perturbation**, which is exactly what it is designed to do.
>
> > Memory usage
>
> Good point! The backward pass is observed to take about twice the amount of FLOPs as a forward pass (https://www.lesswrong.com/posts/fnjKpBoWJXcSDwhZk/what-s-the-backward-forward-flop-ratio-for-neural-networks), and we performed an experiment, finding that a forward+backward pass has about twice the peak memory of a forward pass. Recall that each iteration of p_mmse consists of a forward+backward pass; each iteration of p_mmse is roughly 3x the number of FLOPs and twice the peak memory, compared to a single iteration of p_mc.
>
> > Figure 2: “The estimation error for low sigma is impossible to see. It may be better to report relative deviations from p_mc”
>
> The estimation error for small sigma is much lower than that for larger sigma, making its distribution concentrated near zero. In the figure, we indeed measure p_robust’s relative deviations from p_mc by computing their log ratio.
>
> > Section 3.2.3: why would $x_i \rightarrow \infty \forall i$?
>
> This characterizes the regime in which the approximation holds, where the logit differences are large, i.e., points that are far away from the decision boundary.
>
> > Will code be publicly available?
>
> Yes!
>
> Thank you again for the review! We hope our response addresses your comments and that you consider increasing the score. Thank you very much!
>
> Sincerely,
>
> Authors

---

### Official Review · Reviewer_BEZM · 2024-03-22

**Q2-1 Originality-Novelty:** 3
**Q2-2 Correctness-Technical Quality:** 3
**Q2-5 Clarity Of Writing:** 2

**Q1 Summary And Contributions:**

In this paper, the authors present an approach to approximate the average case robustness of data point for deep neural networks. Typical approaches that look at adversarial robustness of a model look at the worst case (i.e. does an adversarial example exist for a given example within an $\epsilon$ ball), but do not look at the average case (i.e. how easy it is to perturb a given example to generate an adversarial example). Monte Carlo estimate for average case robustness are expensive to compute given that it requires generating a large number of samples owning to the high dimensionality of the features. The authors present different approaches to approximate the average case robustness using linearization and present an empirical study on the quality of their approximate methods when compared to the Monte Carlo estimate.

**Q2-3 Extent To Which Claims Are Supported By Evidence:**

2: Fair: the main claims are somewhat supported by evidence (but the experimental evaluation may be weak, or does not match entirely with the claims, important baselines may be missing, proofs contain important ideas but lack rigor, algorithmic details are only discussed superficially, references are imprecise, assumptions are not sufficiently motivated or explicated, etc.).

**Q2-4 Reproducibility:**

1: Poor: key details (e.g. proof sketches, experimental setup) are incomplete/unclear, or key resources (e.g. proofs, code, data) are unavailable.

**Q3 Main Strengths:**

- The overall idea is well motivated and of great interest to the UAI community. Getting average case robustness gives a good idea of model reliability, especially for practical applications.

- The authors present non-trivial theoretical results when it comes to linearization of non-linear models and provide a range of estimators built on top of them.

**Q4 Main Weakness:**

- The paper is extremely hard to follow. While I appreciate the theoretical details of the paper, the authors should work on improving the presentation of the paper to make it more understandable.

- The implementation of these different estimators is not clear. From the results at least it seems that the approaches would be good for practitioners, but no details on how to implement these are available.

**Q5 Detailed Comments To The Authors:**

- The authors should definitely include an algorithm block and provide pseudocode where possible for each estimator.

- On Page 3, under sec 3.2 it says "If they are orthogonal, then this problem is avoided, as the probability of crossing $y_{12}$ and $y_{13}$ are independent random variables." The rationale behind this is not clear. What if we move towards the intersection of these two hyperplanes? Can we not cross both at the same time then?

- Page 3, notation: What do you mean by expanding order-t tensor to order-(t+1). Also what are these $a_i$'s?

- Lemma 1. Why not specify $t$ in the notation for $p_{\sigma}^{robust}(\mathbf{x})$? This can be misleading otherwise.

- How do we compute $\{g}_i$ in practice? Especially for deep neural networks, it would be non trivial to get a functional form.

- In proposition 1, what is $A_i$ exactly? And how do we compute it for neural networks?

- For Figure 2, do we compute the average of the log ratios over the entire test set? This would be computationally expensive though, for Monte Carlo estimate at least, right? And how do we average over classes?

**Q9 Complying With Reviewing Instructions:**

Yes

---

> ### Author Rebuttal · Authors · 2024-04-04
>
> Dear Reviewer BEZM,
>
> Thank you! We really appreciate your review and are glad you found that our work is “well motivated and of great interest to the UAI community”, that our “theoretical results” are “non-trivial”, and that “average case robustness gives a good idea of model reliability, especially for practical applications”. We address your comments below.
>
> > “From the results at least it seems that the approaches would be good for practitioners, but no details on how to implement these are available.”
>
> In the paper, we provide the mathematical equation for each method. We will also make the code for our experiments, which contains implementations of each method, publicly available.
>
> > "If they are orthogonal, then this problem is avoided, as the probability of crossing y12 and y13 are independent random variables." The rationale behind this is not clear. What if we move towards the intersection of these two hyperplanes? Can we not cross both at the same time then?
>
> Yes, it is indeed possible to cross both at the same time, but these are statistically independent events when the decision boundaries are orthogonal, i.e., the fact that we have crossed one decision boundary, say y12, does not influence the probably of the other decision boundary y13 also being crossed. However, this is not the case if the angle between the decision boundaries is small, as crossing one decision boundary indicates with high probability (under Gaussian noise) that the other decision boundary is also crossed.
>
> > Page 3, notation: What do you mean by expanding order-t tensor to order-(t+1). Also what are these a_i's?
>
> This notation writes an order(t+1) tensor as an order-t tensor by collapsing the former tensor along the class dimension. So, to get the full expression, we need to expand the condensed order-t tensor to the full order-(t+1) tensor. In the “Notation” section, the a_i’s are example terms in the class dimension that are collapsed in the notation.
>
> For example, this notation means that, in Lemma 1 and Definition 2, the term inside \Phi() is not a scalar, but a (C-1)-dimensional vector. We opted for this notation because, otherwise, the equations would be too long to fit in a single column.
>
> > Lemma 1. Why not specify t in the notation for p_robust? This can be misleading otherwise.
>
> When we first introduce p_robust (Definition 1), we include t in the notation. Later (page 3, paragraph 1), we explain that, “In this paper, given that robustness is always measured with respect to the predicted class t at x, we henceforth suppress the dependence on t in the notation”.
>
> > How do we compute g_i in practice? Especially for deep neural networks, it would be non trivial to get a functional form.
>
> The boundary functions g_i(x) are defined as g_i(x) = f_t(x) - f_i(x) for i \neq t, where i is the class index, t is the target class, and f is the model logits (page 4, line 1). Thus, the set of g_i(x)’s is computed by taking the pairwise difference between the logit of the target class and the logits of each of the other classes.
>
> > In proposition 1, what is A_i exactly? And how do we compute it for neural networks?
>
> Please note that Proposition 1 is a theoretical result for models with quadratic decision boundaries, and not neural networks. This quadratic decision boundary is parameterized by A_i, u_i and c, as indicated in the statement on Proposition 1.
>
> > For Figure 2, do we compute the average of the log ratios over the entire test set? This would be computationally expensive though, for Monte Carlo estimate at least, right? And how do we average over classes?
>
> Thank you for the question! Perhaps there is a misunderstanding. Figure 2 evaluates the accuracy of the analytical estimators. For each data point, we compute p_mc and p_analytical (using different analytical estimators). While it’s true that p_mc is computationally expensive, we are simply using it as an estimate of the ground truth. We compare how close p_analytical is to p_mc using their log ratio. We do not compute the “average of the log ratios” nor do we “average over classes”; instead we plot the log ratio for each point, resulting in one distribution (boxplot) for each estimator.
>
> Thank you again for the review! We hope our response addresses your comments and that you consider increasing the score. Thank you very much!
>
> Sincerely,
>
> Authors

---

### Official Review · Reviewer_SwxZ · 2024-03-22

**Q2-1 Originality-Novelty:** 3
**Q2-2 Correctness-Technical Quality:** 4
**Q2-5 Clarity Of Writing:** 4

**Q1 Summary And Contributions:**

This paper introduce novel estimators for average robustness that surpass the efficiency of Monte Carlo estimators.
Also, establish error bounds for these estimators, both for linear and non-linear models, providing a rigorous framework for their application. Futhermore, the paper validate the efficiency and accuracy of the estimators on Deep Neural Networks, with findings suggesting that reduced noise levels correspond to increased accuracy in estimations. Conversely, a more robust model demonstrates higher accuracy in estimations even with larger noise levels. And the paper proves application of these methods involves evaluating the average robustness of individual data points and assessing model bias. Data points with low robustness may resemble support vectors, while those with high robustness may represent canonical examples within the dataset.

**Q2-3 Extent To Which Claims Are Supported By Evidence:**

4: Excellent: all claims are supported by very convincing evidence (in the form of comprehensive experimental evaluation, rigorous mathematical proofs, detailed (pseudo-)code, precise references, well-motivated and realistic assumptions) and the authors deliver what they promise.

**Q2-4 Reproducibility:**

3: Good: key resources (e.g. proofs, code, data) are available and key details (e.g. proofs, experimental setup) are sufficiently well-described for competent researchers to confidently reproduce the main results.

**Q3 Main Strengths:**

This paper presents significant theoretical advancements, including the introduction of three distinct estimators for average robustness. Additionally, offer an approximation for the mvn-cdf. Each approximation is accompanied by error bounds, supported by empirical evidence to validate their accuracy. The empirical findings demonstrate robustness and extendibility to practical applications. Through rigorous experimentation, the theoretical frameworks are validated, showcasing their efficacy in real-world scenarios.

**Q4 Main Weakness:**

This paper is dense with abbreviations, which can pose challenges for readers' comprehension.

**Q5 Detailed Comments To The Authors:**

In this paper, the author points out the necessity of the equation UU^T = I in the approximation of the mvn-cdf. However, such cases are uncommon in real-world scenarios. This discrepancy warrants further explanation.
Furthermore, in Supplementary Figures 5 and 6, the details are difficult to discern, the author probably need to convert them to vector graphics format for improved clarity.

**Q9 Complying With Reviewing Instructions:**

Yes

---

> ### Author Rebuttal · Authors · 2024-04-04
>
> Dear Reviewer SwxZ,
>
> Thank you! We really appreciate your review and are glad you liked our paper! We address your comments below.
>
> > “the author points out the necessity of the equation UU^T = I in the approximation of the mvn-cdf. However, such cases are uncommon in real-world scenarios”
>
> Thank you for the question! This assumption is mainly for analytical and computational tractability, resulting in a closed-form expression for mvn-cdf. We agree that this is uncommon in real-world scenarios. Empirically, we find that, even with the assumption, the analytical estimators accurately estimate p_robust, especially for small values of sigma (Figure 2).
>
> > “in Supplementary Figures 5 and 6, the details are difficult to discern, the author probably need to convert them to vector graphics format for improved clarity”
>
> We apologize about this and will reformat these figures.
>
> Thank you again for the review!
>
> Sincerely,
>
> Authors

---

### Official Review · Reviewer_gT8i · 2024-03-23

**Q2-1 Originality-Novelty:** 3
**Q2-2 Correctness-Technical Quality:** 3
**Q2-5 Clarity Of Writing:** 3

**Q1 Summary And Contributions:**

This paper proposes a novel approach to measuring the robustness of machine learning models, introducing the concept of average-case robustness. This metric aims to offer a more nuanced understanding of model behavior by evaluating the fraction of points in a local region that yield consistent predictions, in contrast to adversarial robustness, which views robustness in binary terms. The authors develop the first analytical estimators for average-case robustness for multi-class classifiers, providing a practical tool for assessing model reliability and identifying data point vulnerabilities.

**Q2-3 Extent To Which Claims Are Supported By Evidence:**

3: Good: the main claims are supported by convincing evidence (in the form of adequate experimental evaluation, proofs, (pseudo-)code, references, assumptions).

**Q2-4 Reproducibility:**

3: Good: key resources (e.g. proofs, code, data) are available and key details (e.g. proofs, experimental setup) are sufficiently well-described for competent researchers to confidently reproduce the main results.

**Q3 Main Strengths:**

- The paper successfully identifies and addresses a gap in existing robustness metrics by proposing average-case robustness, a complementary framework that captures the degree of vulnerability beyond the binary perspective of adversarial robustness.
- The paper develops the first analytical estimators for the average-case robustness of multi-class classifiers, which is an important technical contribution, enabling efficient and insightful robustness evaluations.
- The methodology is tested across various datasets and models, showcasing its versatility and potential for widespread applications.

**Q4 Main Weakness:**

- The scopt of the problem setting is limited. The exploration of robustness primarily revolves around Gaussian noise. The real-world applicability of these findings might be constrained given the myriad of perturbations or corruptions encountered outside controlled, Gaussian-noise scenarios.
- While the so-called average-case robustness is insightful from the perspective of \epsilon-ball based adversarial robustness, the concept is actually equivalent to robustness to random perturbations or corruptions. In this sense, the problem setting is not very appealing.

**Q5 Detailed Comments To The Authors:**

- The paper asserts that "The smaller this fraction, the easier it is to find a misclassified example." However, the ease of identifying misclassified examples is significantly influenced by the optimization method used to search adversarial perturbations. It's not very useful to gauge the level of ease solely based on the fraction mentioned.
- If average-case robustness is delineated as per Definition 1 within the paper, then its value will remain below one. For a non-trivial classifier, there must exist one point $x'$ such that $f(x')\neq t$. For any $x$, it is always possible that $x+\epsilon=x'$. Thus, the average-case robustness will be less than 100%. In this sense, Definition 1 seems to not generalize the classic $\ell_p$ adversarial robustness. Could you explain this?

**Q9 Complying With Reviewing Instructions:**

Yes

---

> ### Author Rebuttal · Authors · 2024-04-04
>
> Dear Reviewer gT8i,
>
> Thank you! We really appreciate your review and are glad you liked the paper! We address your comments below.
>
> > “The real-world applicability of these findings might be constrained given the myriad of perturbations or corruptions encountered outside controlled, Gaussian-noise scenarios.”
>
> While the paper focuses on the case of Gaussian noise, the estimators equally apply to other noise types in the high-dimensional setting as long as they are coordinate-wise independent (Lemma 2). As the first paper to develop analytical estimators of average-case robustness, this paper focuses on the case of Gaussian noise because it provides analytical tractability, and we hope to extend these results to more complex noise types in future work.
>
> > “ease of identifying misclassified examples is significantly influenced by the optimization method used to search adversarial perturbations.”
>
> This is a good point! It is true that, in addition to the size of the misclassified region, another factor that affects the ease of finding misclassified examples is the specific optimization method used. In this study, we aim to study model robustness in a manner agnostic to the specific optimization used, and thus, we only focus on the size of the misclassified region. We believe our study can form the basis for future studies looking into the properties (e.g., “ease of identifying misclassified examples”) of specific optimization methods. We will add this discussion to the paper.
>
> > “Definition 1 seems to not generalize the classic l_p adversarial robustness. Could you explain this?”
>
> Thank you for the question! This is true. If the domain of the underlying noise is bounded (for example, by an l_p ball), then Definition 1 generalizes the classic l_p robustness. If the domain of the underlying noise is not bounded, then Definition 1 does not generalize classical l_p robustness. However, in high dimensions, the unbounded Gaussian ball approaches a bounded uniform distribution on the sphere. Additional details are discussed in Section 3.1 (page 3, 2nd and 3rd paragraph).
>
> Thank you again for the review! We hope our response addresses your comments and that you consider increasing the score. Thank you very much!
>
> Sincerely,
>
> Authors

---

### Official Review · Reviewer_kQcg · 2024-03-25

**Q2-1 Originality-Novelty:** 3
**Q2-2 Correctness-Technical Quality:** 3
**Q2-5 Clarity Of Writing:** 3

**Q10 Ethical Concerns:**

nan

**Q1 Summary And Contributions:**

This paper introduces a novel approach to assessing the robustness of multi-class machine learning models, complementing the traditional binary view of adversarial robustness. The authors propose analytical estimators for average-case robustness, which measure the consistency of predictions across a local region. These estimators are effective for deep learning models and help identify vulnerable data points and quantify model bias. The work enhances our understanding of model behavior in real-world settings.

**Q2-3 Extent To Which Claims Are Supported By Evidence:**

3: Good: the main claims are supported by convincing evidence (in the form of adequate experimental evaluation, proofs, (pseudo-)code, references, assumptions).

**Q2-4 Reproducibility:**

3: Good: key resources (e.g. proofs, code, data) are available and key details (e.g. proofs, experimental setup) are sufficiently well-described for competent researchers to confidently reproduce the main results.

**Q3 Main Strengths:**

1）By introducing the concept of average-case robustness, the work provides a complementary perspective to the traditional binary view of adversarial robustness. This can lead to a more nuanced understanding of model vulnerabilities.
2）Paper provides theoretical support for its claims.
3)  The paper provides empirical evidence to support the effectiveness of the proposed estimators, demonstrating their utility in real-world scenarios.

**Q4 Main Weakness:**

Larger scale datasets are needed to more thoroughly validate the effectiveness of the proposed method.

**Q5 Detailed Comments To The Authors:**

It is better to conduct more experiments on larget scale dataset, such as NICO, WILDs, to verify the method.

**Q9 Complying With Reviewing Instructions:**

Yes

---

> ### Author Rebuttal · Authors · 2024-04-04
>
> Dear Reviewer kQcg,
>
> We thank you for your review and are glad you liked the paper!
>
> Regarding the choice of datasets, we chose a small-scale dataset to help us better perform exhaustive, controlled experiments and conduct a clean experimental study. One computational bottleneck for experiments is the computation of the baseline p_mc, which is intractable for large models and datasets. In practice, we find the number of classes to be another bottleneck (more so than the model and dataset size). For example, calculating p_robust analytically for a model with 1000 classes requires computing a 999-dimensional MVN CDF, which is intractable. This is due to the inherent complexity of the underlying mathematical object (mvn-cdf), which does not have a closed-form solution. However, we hope to extend our work to large-scale models and datasets in future work.
>
> Thank you again for the review!
>
> Sincerely,
>
> Authors

---

### Meta-Review · Area_Chair_ynaq · 2024-04-23

Coming soon.